# Identification of Underground Artificial Cavities Based on the Bayesian Convolutional Neural Network

**DOI:** 10.3390/s23198169

**Published:** 2023-09-29

**Authors:** Jigen Xia, Ronghua Peng, Zhiqiang Li, Junyi Li, Yizhuo He, Gang Li

**Affiliations:** 1School of Geophysics and Geomatics, China University of Geosciences, Wuhan 430074, China; xiajg@crirp.ac.cn; 2The 22nd Research Institute of China Electronics Technology Group Corporation, Xinxiang 453003, China; lizhiqiang316@126.com; 3College of Instrumentation and Electrical Engineering, Jilin University, Changchun 130012, China; heyz20@mails.jlu.edu.cn (Y.H.); ligang2013@jlu.edu.cn (G.L.)

**Keywords:** artificial cavities, identification methods, apparent resistivity imaging method, BCNN

## Abstract

The development of underground artificial cavities plays an important role in the exploitation of urban spatial resources. As the rapidly growing number of underground artificial cavities with different depths and scales increases, the detection and identification of underground artificial cavities has become a key issue in underground engineering studies. Geophysical techniques have been widely used for the construction, management, and maintenance of underground artificial cavities. In this study, we present two identification methods for underground artificial cavities. Apparent resistivity imaging is the most popular technique for quickly identifying underground artificial cavities, using the forward simulation results of a three-dimensional earth model and comparing these with the preset positions of artificial cavities, as demonstrated in the experiment. To further improve the efficiency of underground artificial cavity identification, we developed a fast recognition approach for underground artificial cavities based on the Bayesian convolutional neural network (BCNN). Compared to a traditional convolutional neural network, the performance of the BCNN method was greatly improved in terms of the classification accuracy and efficiency of identifying underground artificial cavities with apparent resistivity image datasets.

## 1. Introduction

In recent years, the construction of underground artificial voids has emerged as a significant trend in urban development, aiming to utilize urban underground space resources. Consequently, the number, depth, and scale of these voids have been rapidly increasing in cities. The application of geophysical exploration methods is crucial for the efficient, accurate, and non-invasive detection and identification of these underground artificial cavities. The expansion of urban underground space, extending from shallow to deep levels, has heightened the importance of studying efficient and precise geophysical exploration technology. This technology is crucial for the development, utilization, and safety management of urban underground space [1,2].

The geophysical electromagnetic method is an exploration technique that analyzes changes or distributions in electromagnetic fields across space and time. It relies on the varying responses of different media to natural or artificial electromagnetic fields. The electromagnetic methods can be categorized as ground-based or airborne based on the application space of detection systems. Ground electromagnetic methods mainly include controlled source audio magnetotelluric methods [3,4,5], wide-field electromagnetic methods [6,7], etc. However, the application of ground-based electromagnetic methods is significantly restricted in areas with complex surface conditions, such as wetlands, deserts, and other locations where conducting ground surveys is logistically challenging. On the other hand, airborne electromagnetic methods can swiftly detect geological structures without being hindered by ground factors. However, the detection depth of the airborne electromagnetic method is relatively shallow due to limitations such as the drone load capacity and launch power [8]. To enhance the detection depth, the semi-airborne electromagnetic method (SAEM) was employed. This technique involves placing high-power electrical sources on the ground and using aerial platforms equipped with acquisition systems to measure electromagnetic responses [9,10,11]. In comparison to airborne electromagnetic methods, semi-airborne methods can measure data with a higher signal-to-noise ratio, resulting in a greater detection depth.

With the emergence of the artificial intelligence era, convolutional neural networks (CNN) in deep learning have received much attention due to their advantages, such as automatic feature learning, a simple structure, and strong adaptability. Consequently, CNNs have found widespread applications across various fields. In the context of underground artificial cavity recognition, CNNs primarily serve target detection and recognition purposes. For example, Reichman et al. [12] employed CNN to detect hazardous underground objects, where CNN utilized B-Scan images as the input and then used GPR images from the test dataset for the detection and recognition of these underground hazards. The results demonstrated a notably high detection rate for underground targets. Dinh et al. [13] employed CNNs for the detection of steel bars within bridge structures, which illustrates the advantages of CNNs in automatic target detection. In addition, many variants of CNNs, including R-CNN [14], Fast-RCNN [15], Faster-RCNN [16,17,18,19], and YOLO [20], have been developed for the detection of underground targets. Pham et al. [21] explored the use of Fast-RCNN for underground target detection. This algorithm involves the training of network parameters using measured and simulated data, followed by the utilization of a trained network model for underground target detection on the test dataset.

In this work, we aim to detect underground artificial cavities using the frequency-domain SAEM method. To this end, we first examine the fundamental SAEM theory and briefly introduce the implementation process of the apparent resistivity imaging method for quick underground artificial cavity identification. The accuracy of this method is then validated through numerical experiments. However, the imaging results using the conventional apparent resistivity imaging method did not automatically identify artificial holes and lacked efficiency when detecting potential underground artificial cavities over a large area. To address this limitation, we developed an artificial hole recognition method based on a Bayesian convolutional neural network [21,22,23]. This approach demonstrates robustness against overfitting and the ability to learn from small datasets. The effectiveness of this proposed method was verified using practical public datasets and COSMOL-simulated artificial cavity imaging data.

## 2. Apparent Resistivity Imaging Method

To quickly image the electromagnetic simulation results and enable the identification of underground artificial cavities, we employed the apparent resistivity imaging method. Vertical magnetic responses in the frequency domain for uniform earth exhibit increased monotonic behavior with resistivity; therefore, solving the inverse function of resistivity allowed us to obtain a distinct and apparent resistivity value. When the resistivity of uniform earth is ρ1, the expression of a vertical magnetic flux density in three-dimensional space is [24]:(1)$$B_{z}=\frac{\mu_{0}IL}{2\pi }\cdot \frac{y}{R}\int_{0}^{\infty }\frac{\lambda^{2}}{\lambda +u_{1}}e^{\lambda z}J_{L}\lambda Rd\lambda$$
where $u_{1}=\sqrt{\lambda^{2}+k_{1}^{2}}$, $k_{1}^{2}=-i\omega \mu /\rho_{1}$. In order to quickly calculate the apparent resistivity in the detection area, the Dichotomy method was used to iteratively calculate the approximate value of apparent resistivity. The calculation process is shown in Figure 1. Before solving the apparent resistivity, it was necessary to first set various measurement line parameters and electrical parameters. The transmitting antenna was 200 m long, and the measurement line was 1000 m long.

The calculation steps for apparent resistivity are as follows:(1)Select the initial resistivity value *ρ*_0_, iteration step size, Δ=2 and allowable error percentage to terminate the iteration ε=0.0001;(2)When resistivity is *ρ*_0_, calculate the corresponding vertical magnetic flux density *B*_z_;(3)Compare the theoretical results and calculated results of vertical magnetic flux density and evaluate whether the current error percentage *ε*_0_ is less than the predefined error *ε*;(4)Terminate the iteration and consider the current resistivity value as the apparent uniform resistivity at the measurement position when the allowable error *ε* is satisfied; otherwise, update the resistivity value by $\rho_{0}=\rho_{0}+\Delta$ and repeat steps (2) and (3);(5)Furthermore, if the vertical magnetic flux density Bz for all resistivity within the predefined range does not meet the allowable error, the resistivity value associated with the smallest error can be chosen as the uniform apparent resistivity.

In addition, when the transmitting and receiving distance exceeds six times that of skin depth, the apparent depth of anomalous bodies detected by the semi-airborne frequency domain electromagnetic method can be directly calculated using the skin depth formula.

## 3. Classification and Recognition Method Based on Bayesian Convolutional Neural Network

By using the apparent resistivity imaging method to obtain a 2D profile image and manually determining whether there was low resistivity in the abnormal body of the image, the identification of underground artificial cavities could be achieved. However, in the actual detection process, a large number of survey lines needed to be detected to obtain multiple 2D apparent resistivity images. In order to effectively and intelligently identify whether underground artificial caves existed in the image, the conventional method used CNNs for recognition processing.

CNNs have demonstrated good performance in processing image data due to their ability to preserve spatial information for each pixel and extract crucial features. The typical structure of a CNN is depicted in Figure 2, and its training process usually consists of the following steps:(1)Data Collection: generate or collect the training samples. The training set should comprise input samples along with their corresponding reference outputs.(2)Network Configuration: determine the number of layers and nodes in each layer of the neural network and select suitable activation and loss functions. These choices depend on the complexity and accuracy requirements of these problems.(3)Initialization: Initialize the weights and biases for each node within the network.(4)Forward propagation: Feed the input samples from the training set through the neural network. Measure the disparity between the model’s output and the reference output using the selected loss function.(5)Backpropagation: Minimize the loss function by adjusting the network’s parameters through the process of backpropagation. This involves calculating gradients and accordingly updating weights and biases.

CNNs iteratively repeat the above steps until their loss value reaches a desired accuracy threshold.

In the CNN structure, the kernel function is an important parameter of the convolutional layer, which is essentially a matrix composed of the weight parameters ω of neurons. The specific process for this is shown in Figure 3. The blue 3 × 4 matrix represents the input image data, the yellow 2 × 2 matrix represents the convolution kernel, and the green matrix represents the feature matrix obtained from the convolution. Due to the fact that, in convolutional operations, data at the edge of the input matrix are only multiplied by the convolutional kernel once, its proportion in the feature matrix is relatively small. However, in some cases, this processing method can lead to the critical information being neglected at the edges of the data.

BCNN is a powerful convolutional neural network architecture. Compared with the traditional convolutional neural network (CNN), BCNN can adequately capture the spatial relationship and global information of images, thus improving the accuracy of image classification, target recognition, and other tasks. Figure 4 illustrates the network structure of the Bayesian convolutional neural network, which is similar to the traditional CNN. However, a key difference lies in how these weights between network nodes are represented. In the Bayesian CNN, these weights are characterized by uncertain probability distributions, whereas in the traditional CNN, they are deterministic values. Within Bayesian convolutional layers, the weight of the convolutional kernel is no longer a single value but a probability distribution, as depicted in Figure 5. It is worth noting that Bayesian convolutional layers retain all the characteristics of traditional convolutional layers. In this work, Gaussian distribution is utilized as the distribution type for each weight. During the training process, the mean and standard deviation of the Gaussian distribution are continuously updated, and the input data of the convolutional layer can be obtained by sampling according to the current probability distribution.

In the backpropagation process, the posterior distribution of each weight in the network can be calculated according to the Bayesian formula, which is expressed as:(2)$$p(\omega |\;D)=\frac{p(D|\;\omega )p(\omega )}{p(D)}$$

Among these p(D|ω)p(ω) is the prior distribution of the weight parameter and the likelihood of probability distribution. However, the calculation of the above posterior probability distribution is often very complex; therefore, Bayesian neural networks usually use variational inference to derive the Loss function. Variational inference assumes that weights follow a posterior ω~q(ω|D) distribution $\omega \sim q_{\theta }(\omega |\;D)$, minimizing the difference between them and their true posterior distribution in the backpropagation process of the neural network, obtaining a probability distribution that can approximate the weight parameters of the network and replace the true distribution. As shown in Figure 6, the loss function of the Bayesian neural network is a problem when minimizing the difference between $q_{\theta }$ and q.

To indicate the degree of consistency between q and $q_{\theta }$, the Kullback–Leibler divergence is introduced, which is expressed as:(3)$$KL[q(\omega )\|\;p(\omega )]=\int q(\omega )\log\frac{q(\omega )}{p(\omega )}d\omega$$

KL divergence can measure the difference between two distributions, and the optimal parameters of the neural network θ are:(4)$$\theta =\underset{\theta }{\mathrm{argmin}}KL\left[q_{\theta }(\omega |\;D)\|\;p(\omega |\;D)\right]$$

$KL\left[q_{\theta }(\omega |\;D)\|\;p(\omega |\;D)\right]$ By unfolding and simplifying, it can be concluded that:(5)$$KL\left[q_{\theta }(\omega |\;D)\|\;p(\omega |\;D)\right]=\log p(D)-\int q_{\theta }(\omega |\;D)\log\frac{p(\omega ,D)}{q_{\theta }(\omega |\;D)}d\theta$$

In Equation (5), logp(D) is referred to as the likelihood of data, which can be regarded as a constant, while the θ optimization process of parameters can be further transformed into the $\int q_{\theta }(\omega |\;D)\log\frac{p(\omega |\;D)}{q_{\theta }(\omega |\;D)}d\theta$ process while minimizing the lower bound of evidence:(6)$$\underset{\theta }{\mathrm{argmin}}KL\left[q_{\theta }(\omega |\;D)\|\;p(\omega )\right]-E_{q(w|\;\theta )}\left[\log_{p}(D|\;\omega )\right]+\log p(D)$$

Compared with Equation (2), although the above equation is also difficult to directly calculate, the loss function of the Bayesian neural network can be expressed by sampling the parameters of the above equation:(7)$$F(D,\theta )=\sum_{i=1}^{n}-\log p\left(D|\;\omega^{\left(i\right)}\right)+\log q_{\theta }\left(\omega^{\left(i\right)}|\;D\right)-\log p\left(\omega^{\left(i\right)}\right)$$

## 4. Numerical Experiments

We first considered a three-dimensional (3D) geoelectrical model to validate the effectiveness of the apparent resistivity imaging method. Then, a large number of simulated apparent resistivity images were drawn as sample datasets to demonstrate the performance of the developed BCNN for the recognition of underground artificial cavities.

The model shown in Figure 7 contains an underground artificial cavity embedded in a homogeneous background. To simulate the SAEM responses of this model, the AC/DC module in the COMSOL software was applied, using a long wire source as the transmitter. Therefore, a magnetic field interface that was simpler than the physical field interface of the electromagnetic field was selected. In addition, frequency domain electromagnetics were selected, and the steady-state solution domain configuration was solved.

An antenna deployed at the surface of the earth was used as the transmitter with a length of 200 m and a current of 50 A. The artificial cavity model was designed using a combination of rectangular and arch structures, and its size was flexible and adjustable considering the actual situations. To simulate the receiving system of SAEM, a measuring line was deployed at a height of 30 m above the ground. The total length of the measuring line was 2000 m, with site spacing at 10 m. In addition, various media properties were set, with the resistivity of air set to 1 × 10^5^ Ω·m, earth’s resistivity set to 500 Ω·m, and the relative dielectric constant and relative permeability set to one.

To compute the SAEM responses, the 3D model was discretized within the Component Geometry of the COMSOL software. The interior of this model is divided into two parts: the upper part is the atmosphere with a size of 6000 m × 4000 m × 1000 m (length × width × height), and the lower part is a large stratum with a size of 6000 m × 4000 m × 1000 m. In addition, for the meshes in the vicinity of the antenna, the cavity and measurement sites are usually refined in order to obtain accurate simulation results. The final discretized model using unstructured tetrahedral meshes is shown in Figure 8. When solving electromagnetic fields, it is usually necessary to establish a domain with open boundaries such that there is no reflection in electromagnetic fields on the boundaries of the computational domain. In the COMSOL software, the Perfect Matching Layer (PML) can be employed as an additional domain outside the model. Essentially, it is just an anisotropic, complex dielectric constant with a magnetic permeability domain that can absorb all electromagnetic waves emitted by the model without any reflection.

The size of the cavity was 30 m × 30 m, which was covered by a layer of concrete wall at a thickness of 3 m and a resistivity of 1 Ω·m. Apparent resistivity imaging calculations were conducted for three different sets of geoelectric models representing underground artificial cavities. The location parameters of the underground artificial cavity and the frequency of the transmitter antenna for each of the three sets of models are shown in Table 1.

The electromagnetic response data obtained from COMSOL simulation software were used to calculate the apparent resistivity and apparent resistivity fitting curve with the apparent conductivity imaging results shown in Figure 9, Figure 10 and Figure 11. Figure 9 demonstrates how the center target of the artificial cavity was located at an offset of −250 m from the center, with a depth of approximately 180 m. The experimental results in the red box were relatively in agreement with the designated position of the artificial cavity. Figure 10 reveals that the center target of the artificial cavity was positioned at an offset of −20 m from the center, with a depth of around 170 m. The experimental results in the red box exhibited a close correspondence to the designated center position. Lastly, Figure 11 illustrates how the center target of the artificial cavity was situated at an offset of −50 m from the center, with a burial depth of roughly 50 m. The experimental results in the red box also closely match the designated center position.

In order to verify the effectiveness of the Bayesian convolutional neural network at recognizing underground artificial cavities, this section employed neural network methods to classify the presence or absence of underground artificial cavities within the sample set, with the aim of achieving the accurate recognition of these artificial cavities.

Using the aforementioned approach, the electromagnetic response data acquired from the forward simulation of 3D geoelectric models were utilized to invert and solve the apparent resistivity images. These simulated apparent resistivity images were then generated to expand the sample set. The sample set consisted of both the simulated apparent resistivity images and added noisy apparent resistivity images. Figure 12 shows partial samples in the sample set. The labels within the sample set were divided into two categories: underground artificial cavities and those without underground artificial cavities. This included 50 sets of apparent resistivity imaging maps and 150 sets of added noisy apparent resistivity imaging maps. In the labeled samples containing underground artificial cavities, the positions and sizes of these cavities were randomized. The samples labeled as not containing underground artificial cavities consisted of images depicting non-anomalous underground bodies as well as randomly positioned and sized low-resistivity anomalous bodies in order to prevent the interference of low-resistivity anomalous bodies that could exist underground, ensuring that the focus remained on the recognition of underground artificial cavities.

From the aforementioned sample set, 150 groups were selected for the training set, while the remaining 50 groups were assigned to the verification set. The AlexNet network structure was employed to compare the training results of the BCNN and the CNN. The learning rate was set to 0.001, the number of training iterations epoch was set to 100, and the batch size was set to 10. Two different loss functions, namely Cross-Entropy Loss and ELBO Loss, were utilized. The accuracy curve during the training process is depicted in Figure 13, and the recognition accuracy is shown in Table 2. The recognition accuracy of underground artificial cavities was reported as 75.05% and 82.77%, respectively. The results demonstrate that both the CNN and the BCNN used in this study yielded favorable outcomes when identifying the presence of underground artificial cavities in apparent resistivity images. However, the BCNN achieved a higher recognition accuracy, thereby enabling a more precise and effective identification of underground artificial cavities.

## 5. Discussions and Conclusions

This study aimed to develop efficient approaches to identify artificial cavities in the ground space using frequency domain semi-airborne electromagnetic responses. We first derived the responses of SAEM in a 1D-layered earth model, which is fundamental for the apparent resistivity imaging method. The effectiveness of this apparent resistivity imaging method was validated on synthetic SAEM data for a 3D geoelectric exploration model simulated by COMSOL software. However, the accuracy of the apparent resistivity imaging method in target detection was dependent upon the complexity of the subsurface and the interpreter’s experiences, which could not automatically identify artificial cavities. In addition, this method lacked efficiency when detecting potential underground artificial cavities over a large area. In order to develop an efficient and intelligent detection and recognition method for underground artificial cavities, we propose the use of a Bayesian convolutional neural network (BCNN). The BCNN is employed to recognize underground artificial cavities by training it on the apparent resistivity image dataset of the preset artificial cavity geoelectric model. The classification results demonstrate the excellent performance of the proposed method in recognizing underground artificial cavities.

In this work, we conducted a detailed study on the identification method of underground artificial cavities based on SAEM responses. However, there are still some limitations to this study, and improvements need to be made in the future to address the following limitations:(1)The model is relatively simple; however, in practical situations, the geological structure is often complex and cannot be approximately simplified as a layered structure. Therefore, it is necessary to establish accurate and detailed geoelectrical models based on actual geological conditions.(2)Although 3D forward modeling results were simulated, the calculation results did not take into account realistic signal noises and other factors that could exist in actual testing.(3)The Bayesian neural network used in this work could recognize whether there were underground artificial cavities in the model after training, but there is still room for improvement in its recognition accuracy. Therefore, further research is needed to improve the recognition accuracy of underground artificial cavities.

## Figures and Tables

**Figure 1 sensors-23-08169-f001:**
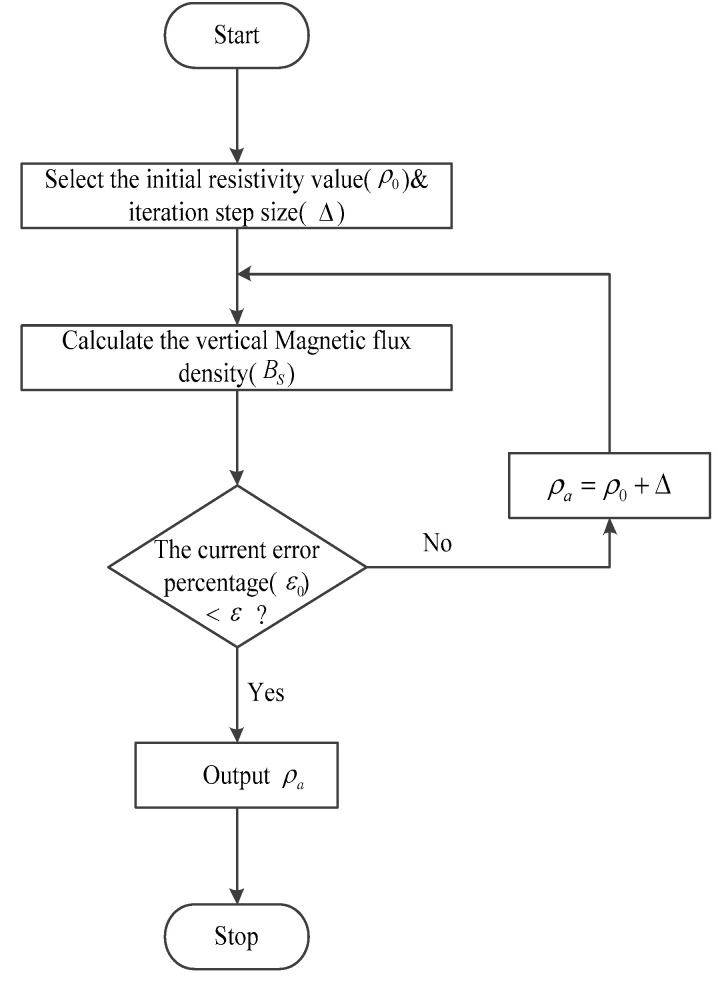
Flow chart for calculating apparent resistivity in the ground space frequency domain.

**Figure 2 sensors-23-08169-f002:**
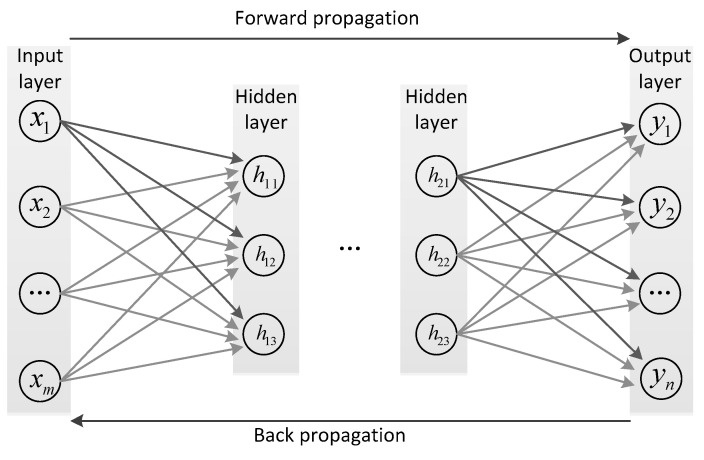
Conventional neural network structure.

**Figure 3 sensors-23-08169-f003:**
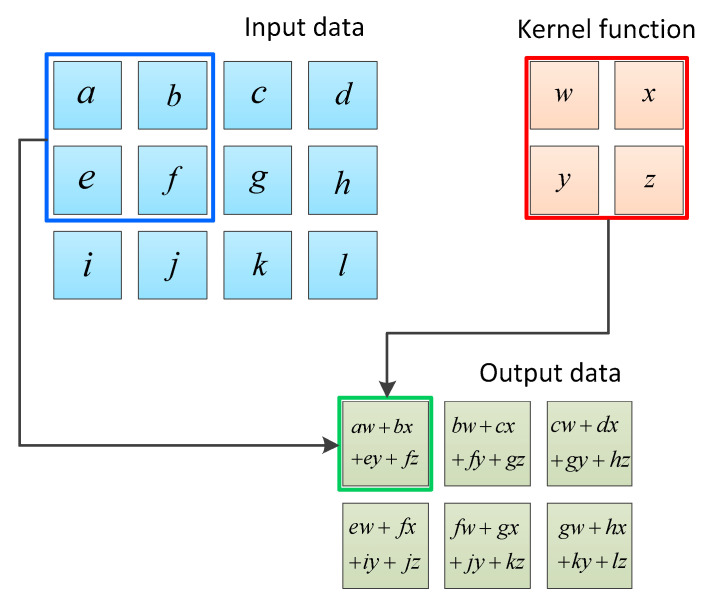
Schematic diagram of convolutional neural network.

**Figure 4 sensors-23-08169-f004:**
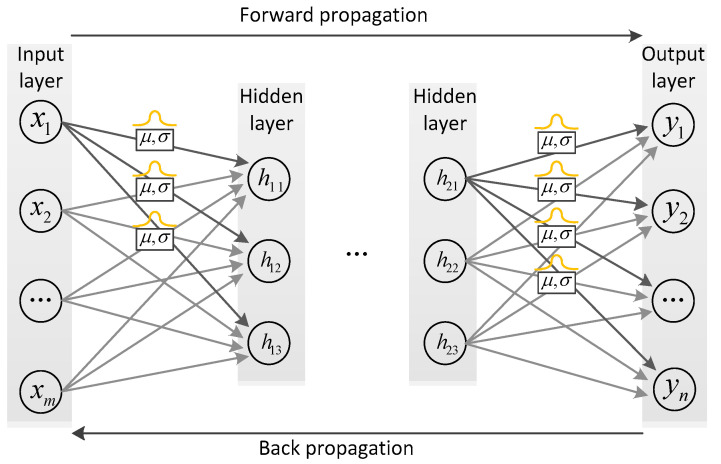
Bayesian neural network structure.

**Figure 5 sensors-23-08169-f005:**
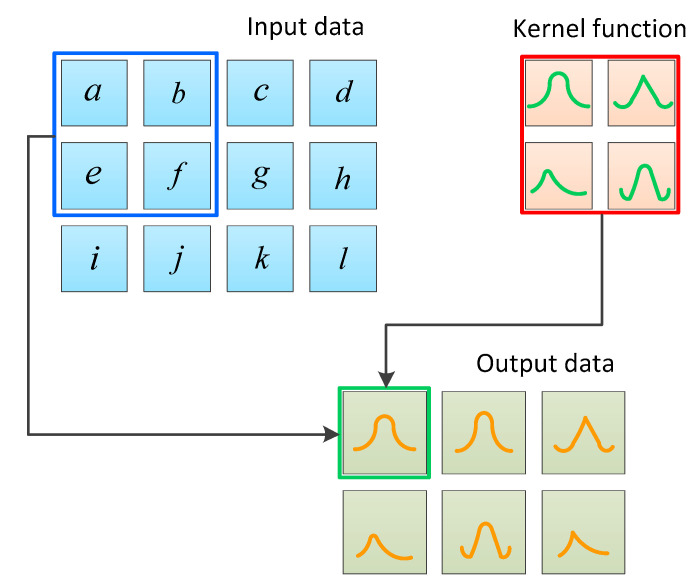
Schematic diagram of Bayesian convolutional neural network.

**Figure 6 sensors-23-08169-f006:**
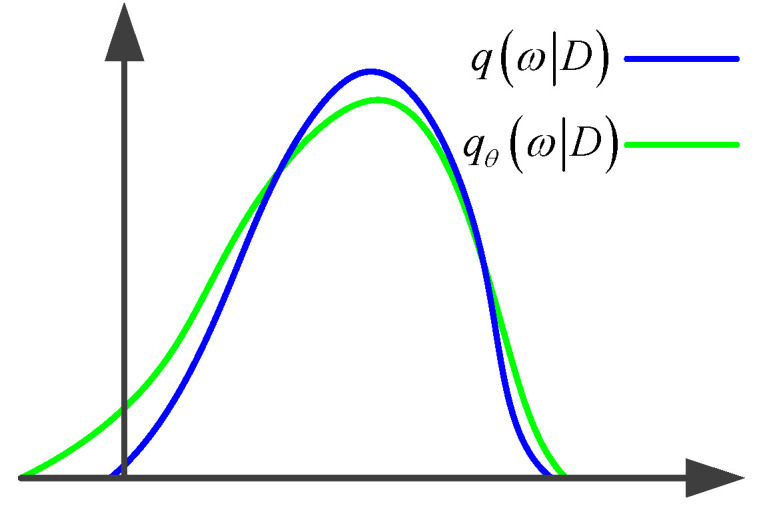
Schematic diagram of variational inference.

**Figure 7 sensors-23-08169-f007:**
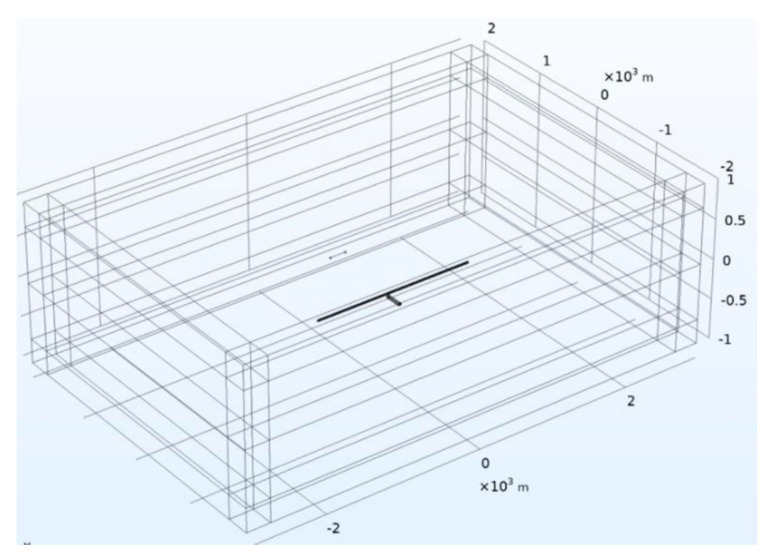
COMSOL simulation model.

**Figure 8 sensors-23-08169-f008:**
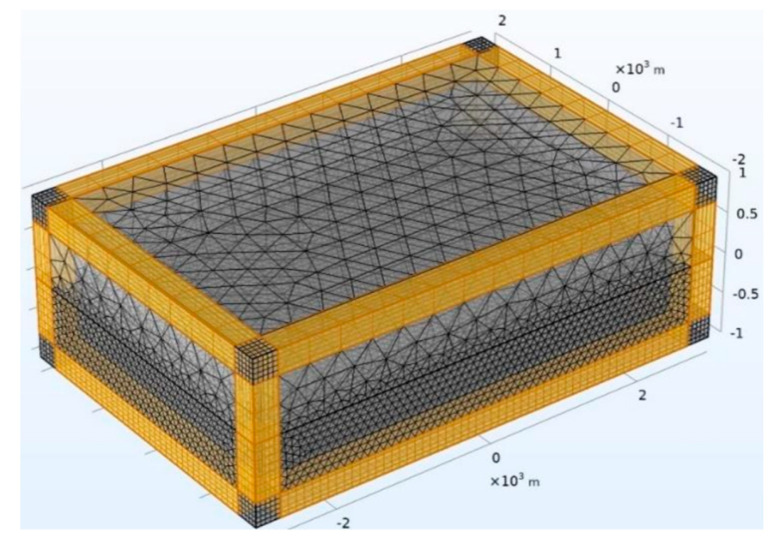
Mesh generation and PML schematic diagram.

**Figure 9 sensors-23-08169-f009:**
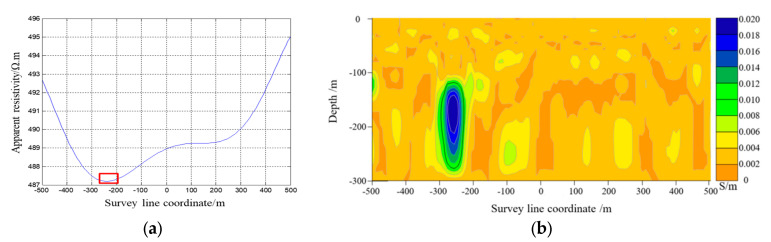
(**a**) Apparent resistivity fitting curve in Experiment 1. (**b**) Contour map of apparent conductivity in Experiment 1.

**Figure 10 sensors-23-08169-f010:**
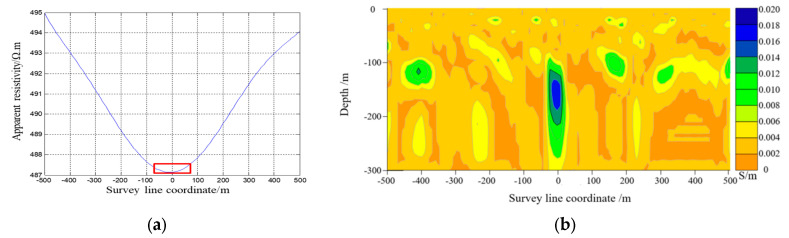
(**a**) Apparent resistivity fitting curve in Experiment 2. (**b**) Contour map of apparent conductivity in Experiment 2.

**Figure 11 sensors-23-08169-f011:**
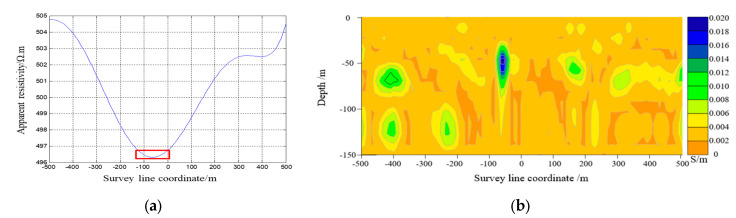
(**a**) Apparent resistivity fitting curve in Experiment 3. (**b**) Contour map of apparent conductivity in Experiment 3.

**Figure 12 sensors-23-08169-f012:**
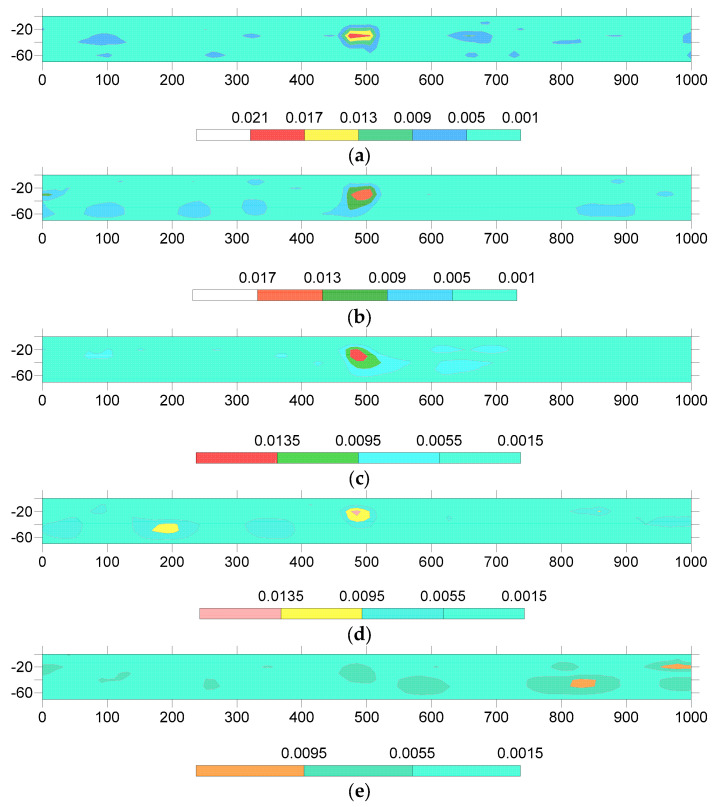
Partial samples in the sample set. (**a**) No noise. (**b**) 10% noise. (**c**) 30% noise. (**d**) 50% noise. (**e**) 70% noise.

**Figure 13 sensors-23-08169-f013:**
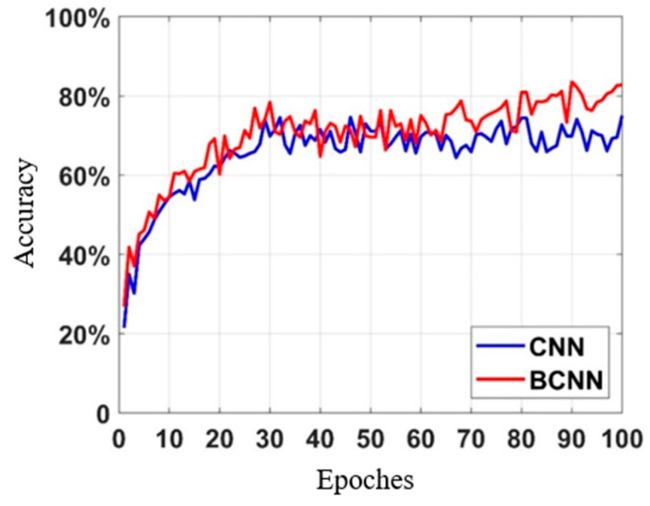
Comparison and recognition results from the underground artificial cavity detection dataset.

**Table 1 sensors-23-08169-t001:** Parameters of apparent resistivity imaging experiment.

	Hollow Burial Depth/m	Void Line Coordinates/m	Emission Frequency/Hz
Experiment 1	200	−200	3200
Experiment 2	200	0	3200
Experiment 3	50	0	25,000

**Table 2 sensors-23-08169-t002:** Classification accuracy of the neural network.

Dataset	Method	Accuracy
Underground artificial	CNN AlexNet	75.05%
Cavity	BCNN AlexNet	82.77%

## Data Availability

All data included in this study are available upon request by contact with the corresponding author.

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
