# Peer review of "Identification of Underground Artificial Cavities Based on the Bayesian Convolutional Neural Network"

_sensors, 2023, doi:10.3390/s23198169_

Round 1

Reviewer 1 Report

How to accurately detect and identify the underground artificial cavities is the key problem of underground engineering research. This paper proposes two identification methods for underground artificial cavities are proposed. In order to further improve the efficiency of underground artificial cavities recognition, the paper developed an underground artificial cavities recognition efficiency based on Bayesian convolutional neural network (BCNN) and compared it with the traditional convolutional neural network. The results of case study show that the Bayesian convolutional neural network method has good performance in the recognition of underground artificial cavities. While the paper is well written and the experimental results are sound in general, I suggest a major revision to the manuscript to consolidate the presented work.

1. Introduction: Lines 56-63, when introducing based on Bayes convolutional neural network, only the advantages of BCNN are introduced. It is recommended to fully describe other methods used in previous studies and the reasons why choosing this model approach. In addition, the literature citations of the article are limited to cover the state of the art

2.The writing of the paper is not sound enough. There is no separate results and conclusions in the paper. It is suggested to optimize the structure of the article, highlight the results of the paper and write the conclusion.

4. Discussion:The connection between experiments 1, 2, and 3 lacks description. It is suggested that the authors reinforce the motivation for setting up the three experiments, making the logic flow better.

5. The paper has some mistakes in writing. Punctuation is not accurate, it is recommended to proofread the whole text. For example, line 72 puts a comma and a period together;

6. The paper gives both Cross-Entropy Loss and ELBO Loss. What is the association between the two losses? Please clarify.

The paper contains errors in its writing, with inaccurate punctuation. It is advisable to have the entire text proofread and improved with the assistance of a native speaker.

Author Response

Thank you very much for taking the time to review this manuscript. We have used english editing services provided by MDPI. Please find the detailed responses below and the corresponding corrections highlighted in the re-submitted files.

Comment 1: Introduction: Lines 56-63, when introducing based on Bayes convolutional neural network, only the advantages of BCNN are introduced. It is recommended to fully describe other methods used in previous studies and the reasons why choosing this model approach. In addition, the literature citations of the article are limited to cover the state of the art.

Response:

Thanks for your comment. We have added the research content of conventional convolutional neural networks and analysis the drawbacks of it, and marked it in red italics in the third part. In addition, for the identification of underground artificial cavities, a reference to convolutional neural networks has been added in the first part with red italic font.

Comment 2: The writing of the paper is not sound enough. There is no separate results and conclusions in the paper. It is suggested to optimize the structure of the article, highlight the results of the paper and write the conclusion.

Response:

Thanks for your comment. We have elaborated on my findings in the results section and provide a clearer exposition of the study's implications and limitations in the summary with red italic font.

Comment 3: The connection between experiments 1, 2, and 3 lacks description. It is suggested that the authors reinforce the motivation for setting up the three experiments, making the logic flow better.

Response:

Thanks for your comment. In the first paragraph of the fourth part, we added a description of the relationship between three experiments, highlighted in red italics.

Comment 4: The paper has some mistakes in writing. Punctuation is not accurate, it is recommended to proofread the whole text. For example, line 72 puts a comma and a period together.

Response:

Following your suggestion, we have proofread the whole text.

Comment 5: The paper gives both Cross-Entropy Loss and ELBO Loss. What is the association between the two losses? Please clarify.

Response:

Cross entropy can measure the difference between the true probability distribution and the predicted probability distribution. ELBO optimizes Bayesian estimation through variational inference. Cross-Entropy Loss and ELBO Loss are two loss functions optimized for CNN and Bayesian optimization.

Reviewer 2 Report

The paper introduce a new method to identify underground artificial cavities,which is more accuracy and efficiency, the contents of the paper is substantial and method is correct, The conclusion is useful for the analysis of underground artificial cavities compare with apparent resistivity imaging method, Therefore The paper need minor revision before accepting.
Comments:
1,The size of the cavities are not mentioned in the paper, the accuracy of different size will be different.
2,what is the basis of The accuracy curve in Fig.9? The accuracy of BCNN become higher while epochs get high.
3,Why choose 50 and 200m case for the burial depth?, how about the other depth.
4,The experiments in the paper are based on the data from simulations which are very ideally, it would be more reliable if this methods tested using dataset from real projects.

no comments for this.

Author Response

Thank you very much for taking the time to review this manuscript. We have used english editing services provided by MDPI. Please find the detailed responses below and the corresponding corrections highlighted in the re-submitted files.

Comment 1: The size of the cavities are not mentioned in the paper, the accuracy of different size will be different.

Response:

Thanks for your comment. The size of the cavities are very important in the experiments. We have added the size of the cavities in the simulation.

Comment 2: What is the basis of The accuracy curve in Fig.9? The accuracy of BCNN become higher while epochs get high.

Response:

The accuracy is the ratio of correctly identifying cavities from the samples. The accuracy of BCNN become higher while epochs get high because the accuracy of high epoch calculations is based on the results of low epoch.

Comment 3: Why choose 50 and 200m case for the burial depth?, how about the other depth.

Response:

Thanks for your suggestion. With the investigation, the Artificial cavities are generally within a range of 50-200 meters underground. So 50 and 200m cases for the burial depth are chosen in this research.

Comment 4: The experiments in the paper are based on the data from simulations which are very ideally, it would be more reliable if this methods tested using dataset from real projects.

Response:

Thanks for your comment. Your suggestion is very insightful. In this paper, the BCNN is proposed to identify the underground artificial cavities. With the simulation results, the model of the artificial cavities are specific. The contour map of apparent conductivity is clear. In the real projects, the results of contour map of apparent conductivity is uncertain, with problems of multiplicity of solutions.

Reviewer 3 Report

The authors have provided a commendable effort in exploring AI applicability. However, there seems to be a noticeable lack in the analysis of previous research. This omission makes it challenging to discern the major academic advancements presented in this work. Given the paper's title, abstract and introduce, it appears that the authors are applying Bayesian Convolutional Neural Networks rather than innovating upon them.

If this is the case, a comprehensive literature review highlighting instances where these networks were both utilized and not utilized would be beneficial. The current introduction, however, falls short in this aspect.

The use of COMSOL simulation software, as mentioned in the main text, warrants a detailed explanation for the benefit of future researchers.

Additionally, the sample size utilized for the CNN application appears to be on the smaller side. The authors should address whether this sample size is sufficient to validate the AI's learning capabilities.

I recommend that the authors elaborate on their findings in the results section and provide a clearer exposition of the study's implications and limitations in the summary.

The authors have provided a commendable effort in exploring AI applicability. However, there seems to be a noticeable lack in the analysis of previous research. This omission makes it challenging to discern the major academic advancements presented in this work. Given the paper's title, abstract and introduce, it appears that the authors are applying Bayesian Convolutional Neural Networks rather than innovating upon them.

If this is the case, a comprehensive literature review highlighting instances where these networks were both utilized and not utilized would be beneficial. The current introduction, however, falls short in this aspect.

The use of COMSOL simulation software, as mentioned in the main text, warrants a detailed explanation for the benefit of future researchers.

Additionally, the sample size utilized for the CNN application appears to be on the smaller side. The authors should address whether this sample size is sufficient to validate the AI's learning capabilities.

I recommend that the authors elaborate on their findings in the results section and provide a clearer exposition of the study's implications and limitations in the summary.

Author Response

Thank you very much for taking the time to review this manuscript. We have used english editing services provided by MDPI. Please find the detailed responses below and the corresponding corrections highlighted in the re-submitted files.

Comment 1: The authors have provided a commendable effort in exploring AI applicability. However, there seems to be a noticeable lack in the analysis of previous research. This omission makes it challenging to discern the major academic advancements presented in this work. Given the paper's title, abstract and introduce, it appears that the authors are applying Bayesian Convolutional Neural Networks rather than innovating upon them. If this is the case, a comprehensive literature review highlighting instances where these networks were both utilized and not utilized would be beneficial. The current introduction, however, falls short in this aspect.

Response:

We really appreciate your efforts in reviewing our manuscript, and thank you for your comments and suggestions. According to your suggestions, we have added a brief literature review about the use of Convolutional Neural Networks for target detection. Please refer to the introduction section in the revised manuscript.

Comment 2: The use of COMSOL simulation software, as mentioned in the main text, warrants a detailed explanation for the benefit of future researchers.

Response:

Thanks for your comment. We have added necessary descriptions about COMSOL modeling in the fourth part of this article, which can provide reference for other researchers. Please refer to the italics highlighted in red for details.

Comment 3: Additionally, the sample size utilized for the CNN application appears to be on the smaller side. The authors should address whether this sample size is sufficient to validate the AI's learning capabilities.

Response:

Thanks for your comment. In the practical detection of the artificial cavities, the accuracy is less than a half because the geophysical detection method has its inherent defects, such as ambiguity, noise, background field. In this paper, the proposed method was verified by simulation results. In the simulation, the performance of the proposed method is good enough in the sample size compared with the common method. And we think the sample size is sufficient to validate the AI’s learning capabilities.

Comment 4: I recommend that the authors elaborate on their findings in the results section and provide a clearer exposition of the study's implications and limitations in the summary.

Response:

Following your suggestion, we have elaborated the findings of the study and provide a description of the limitations of the developed BCNN methods. Please refer to the conclusion section in the revised manuscript.

Round 2

Reviewer 3 Report

I recommend that this manuscript be enrolled in sensors journal.

I recommend that this manuscript be enrolled in sensors journal.